# Differentiable Boundary Point Extraction for Weakly Supervised Star-shaped Object Segmentation

**Robin Camarasa**[1,2*]                                    R.CAMARASA@ERASMUSMC.NL
**Hoel Kervadec**[1,2]                                      H.KERVADEC@ERASMUSMC.NL
**Daniel Bos**[2,3]                                         D.BOS@ERASMUSMC.NL
**Marleen de Bruijne**[1,2,4]                    MARLEEN.DEBRUIJNE@ERASMUSMC.NL

[1] *Biomedical Imaging Group Rotterdam, Erasmus MC, Rotterdam, The Netherlands*

[2] *Department of Radiology and Nuclear Medicine, Erasmus MC, Rotterdam, The Netherlands*

[3] *Department of Epidemiology, Erasmus MC, Rotterdam, The Netherlands*

[4] *Department of Computer Science, University of Copenhagen, Denmark*

**Editors:** Under Review for MIDL 2022

## Abstract

Although Deep Learning is the new gold standard in medical image segmentation, the annotation burden limits its expansion to clinical practice. We also observe a mismatch between annotations required by deep learning methods designed with pixel-wise optimization in mind and clinically relevant annotations designed for biomarkers extraction (diameters, counts, etc.). Our study proposes a first step toward bridging this gap, optimizing vessel segmentation based on its diameter annotations. To do so we propose to extract boundary points from a star-shaped segmentation in a differentiable manner. This differentiable extraction allows reducing annotation burden as instead of the pixel-wise segmentation only the two annotated points required for diameter measurement are used for training the model. Our experiments show that training based on diameter is efficient; produces state-of-the-art weakly supervised segmentation; and performs reasonably compared to full supervision.

Our code is publicly available:
[https://gitlab.com/radiology/aim/carotid-artery-image-analysis/diameter-learning](https://gitlab.com/radiology/aim/carotid-artery-image-analysis/diameter-learning)

**Keywords:** Image segmentation ; weak annotations ; Carotid artery stenosis

## 1. Introduction

In the recent years, huge steps have been made in the direction of automated medical image segmentation as a consequence of, on the one hand, the development of new and powerful deep learning models, and, on the other hand, the increasing availability of open access to large, high-quality (e.g. fully segmented and accurate) datasets for model training. However, as each task is unique, data-scarcity remains an issue for many applications. In most applications full segmentations are not made during clinical practice and therefore are rarely available. However, in many cases, simpler annotations are made by the clinicians, not with a segmentation goal in mind, but rather to estimate the relevant imaging bio-markers.

---

* Corresponding author

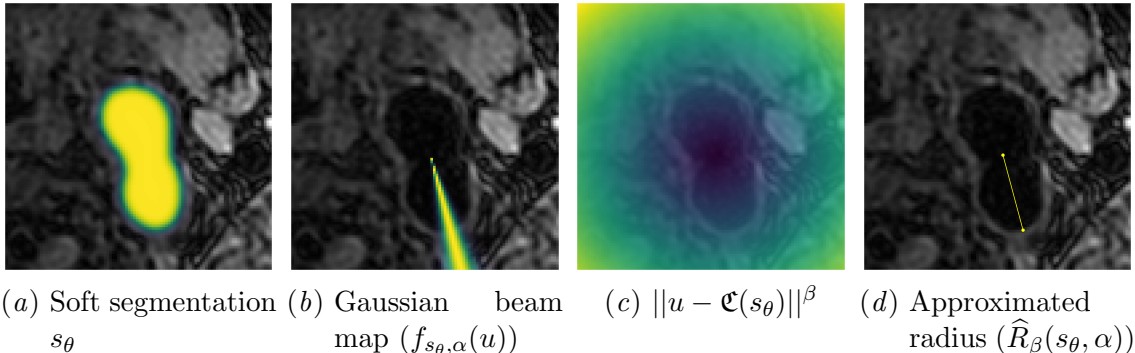

(*a*) Soft segmentation $s_\theta$    (*b*) Gaussian beam map $(f_{s_\theta,\alpha}(u))$    (*c*) $||u - \mathfrak{C}(s_\theta)||^\beta$    (*d*) Approximated radius $(\widehat{R}_\beta(s_\theta, \alpha))$

Figure 1: Overview of the different steps of our method—overlayed on an MRI—going from continuous probabilities (a) to a boundary point (d). Best viewed in colors.

For instance, in the case of carotid artery atherosclerosis[1], clinicians measure the diameter of the lumen at different places in the artery, enabling them to quickly assess the degree of stenosis (i.e. the local *narrowing* of the artery, due to the wall *thickening*). Although, from a clinical perspective it is widely acknowledged that a full segmentation of the lumen would be more informative, and could enable more thorough analysis, it is too time consuming to put in practice.

Yet, the wealth of existing diameter measurements, could be used to train a segmentation network which could ultimately provide full-lumen segmentation and allow to train models on larger and more varied datasets without extra annotation costs. This type of task-specific weak label has not, to the best of our knowledge, been used in the past. As such, existing and popular weakly supervised methods cannot be used directly and a new and efficient way to use those annotations is required.

Our contributions can be summarized as follows:

- we propose a differentiable module, that for a given angle measure the radius of a predicted segmentation;

- we then comprehensively evaluate the proposed method with 3 runs of 4-fold cross-validation and compare to the relevant existing methods from the literature.

## 2. Related work

Weakly supervised segmentation has received an increased attention in the recent years, and most methods use "standard" labels: bounding boxes (Papandreou et al., 2015; Rajchl et al., 2016; Kervadec et al., 2020), scribbles (Lin et al., 2016; Wang et al., 2018), dots (Bearman et al., 2016; Qu et al., 2019; Luo et al., 2021; Dorent et al., 2021) and even classification labels (image tags) (Pathak et al., 2015; Kervadec et al., 2019; Sahasrabudhe

---

1. The thickening of the vessel wall of the carotid artery, an important risk factor for ischemic stroke (World Health Organization, 2011) and a leading cause of death and disability worldwide (World Health Organization, 2014).

et al., 2020). Most of those methods are "generic", in the sense that they can be applied to a variety of tasks without further modification.

However, when strong priors on the segmented shape exists, some weakly supervised methods can be tailored to efficiently exploit this information and compensate the lack of full-annotations. For instance, Sahasrabudhe et al. (2020) and Qu et al. (2019) exploit the structure of deep nuclei segmentation by either learning the image zoom-levels (as a proxy-task), or building Voronoi cells from the dot annotations, respectively. Dorent et al. (2021) assumes that the object is continuous and "draw" a path between the extreme points annotations, by minimizing a geodesic distance. In a multi-instance object detection problem, Yang et al. (2020) detects circles using center and radius information at train time, which can be seen as a segmentation with a strong circular prior.

Two existing methods could be tweaked for the aforementioned task of carotid artery segmentation, when using only a diameter as label. InExtremIS (Dorent et al., 2021) can use the diameter as a valid path without resorting to the complex geodesic distance calculation; but it would not exploit the diameter information. CircleNet (Yang et al., 2020) can be modified to predict a single circular segmentation, using the diameter information but ignoring the exact boundary of the object (as it simply predicts a circle). As such, to the best of our knowledge, there exists no method in the current literature that could use the maximum diameter information while segmenting the object precisely.

## 3. Method

Let us denote[2] an input image $I : \Omega \to \mathbb{R}$, with $\Omega \in \mathbb{R}^2$ a 2D image space, for which we want to predict its corresponding segmentation $y : \Omega \to \{0, 1\}$. Due to the supervision setting, we do not have access to $y$ at training time, and must instead rely on the diameter annotations: $(l, l') \in \Omega^2$, which are simply two points placed on the object boundary. Those points enable us to deduce more information about the object: i) the location of its centroid, ii) its maximum diameter.

To achieve our goal of segmenting the carotid artery, we proceed in different steps: 3.1) locating the centroid of the segmentation, 3.2) extracting points on the boundary of the predicted segmentation in a differentiable way, 3.3) deriving the diameter of the predicted segmentation, and 3.4) creating a combined loss using the aforementioned modules. The first three steps are illustrated in Figure 2.

### 3.1. Location of the vessel

As multiple blood vessels can be present in a single image, the first step is to locate the segmented artery; to "anchor" the segmentation in the correct place during training. As Kervadec et al. (2021) did, the centroid can be efficiently computed as a probability-weighted average of pixel coordinates[3]:

$$\mathfrak{C}(s_\theta) = \frac{1}{\int_{u \in \Omega} s_\theta(u) \, \mathrm{d}u} \int_{u \in \Omega} u.s_\theta(u) \, \mathrm{d}u, \tag{1}$$

2. For convenience, all mathematical notation is summarized in Appendix A, though each symbol is introduced in time in the manuscript.

3. In this paper, we use an integral notation, for consistency with the rest of the formulation. The implementation will eventually discretize it as a sum.

where $s_\theta : \Omega \to [0,1]$ are the predicted probabilities. $\mathfrak{C}(s_\theta)$ is illustrated in Figure 2(a).

## 3.2. Boundary points extraction

With the centroid information, we can derive the boundary coordinates of the segmentation, for different angles $\alpha \in [0, 2\pi[$. This is done in several sub-steps.

First, a Gaussian beam map (shown in Figure 1(b)) restricts the computation to a continuous area around the angle $\alpha$:

$$f_{s_\theta, \alpha}(u) = g_\alpha \left( \Phi_{s_\theta}(u)_\gamma \right) = \frac{1}{\sigma \sqrt{2\pi}} \exp \left( \frac{\Phi_{s_\theta}(u)_\gamma - \alpha}{2\sigma^2} \right), \tag{2}$$

where $\Phi_{s_\theta} : \Omega \to \mathbb{R}^+ \times [0, 2\pi[$ is a function to go from Cartesian to polar coordinates, $\sigma$ a hyper-parameter, controlling the "width" of the beam. $\Phi_{s_\theta}(u)_\gamma$ is the angular coordinate. The Gaussian beam-map can then be embedded into a weighted *average distance to the centroid* of the soft segmentation:

$$\mathcal{I}_\beta(s_\theta, \alpha) = \int_{u \in \Omega} \underbrace{f_{s_\theta, \alpha}(u)}_{\text{beam}} s_\theta(u) \underbrace{||u - \mathfrak{C}(s_\theta)||_2^\beta}_{\text{dist. to centroid}} du, \tag{3}$$

where $\beta$ is an exponent balancing the contribution of the *distance to the centroid*. It can be shown—and this is the main theoretical result of this paper, see Appendix B—that this weighted average is a *differentiable* approximation of the exact radius:

$$\mathcal{I}_\beta(s_\theta, \alpha) \approx \frac{\mathfrak{R}(s_\theta, \alpha)^{\beta+2}}{\beta + 2}. \tag{4}$$

This enables the definition of a first approximate radius function $\widehat{\mathfrak{R}}_\beta$:

$$\widehat{\mathfrak{R}}_\beta(s_\theta, \alpha) = ((\beta + 2)\, \mathcal{I}_\beta(s_\theta, \alpha))^{\frac{1}{\beta+2}}. \tag{5}$$

Notice that this approximate still depends on a parameter $\beta$. As multiple soft segmentations would result in the same radius estimate for a single $\beta$, we average different radius estimates over several $\beta$ to obtain a stronger final radius estimation as:

$$\widehat{\mathfrak{R}}(s_\theta, \alpha) = \mathbb{E}_{\beta \in B} \left( \widehat{\mathfrak{R}}_\beta(s_\theta, \alpha) \right), \tag{6}$$

with $B \subset \mathbb{R}^+$ a discrete set of $\beta$. Finally, we can go back to the Cartesian space with the function $\Phi_{s_\theta}^{-1}(\cdot, \alpha) : \mathbb{R}^+ \times [0, 2\pi[ \to \Omega$ that reverses the initial coordinates transform $\Phi_{s_\theta}$ for the angle $\alpha$ used in Eq. (2). Notice that this radius estimation is fully differentiable, and as such, can be back-propagated over.

## 3.3. Diameter extraction

The diameter estimate is then defined as the maximum distance between the different boundary coordinates, for a discrete set $\Gamma$ of angles:

$$\widehat{\mathfrak{D}}(s_\theta) = \max \left\{ \left|\left| \Phi_{s_\theta}^{-1} \left( \widehat{\mathfrak{R}}(s_\theta, \alpha), \alpha \right) - \Phi_{s_\theta}^{-1} \left( \widehat{\mathfrak{R}}(s_\theta, \alpha'), \alpha' \right) \right|\right|_2, \tag{7}$$

$$\forall \left( \alpha, \alpha' \right) \in \Gamma^2 \right\}.$$

### 3.4. Losses, regularizer, and final model

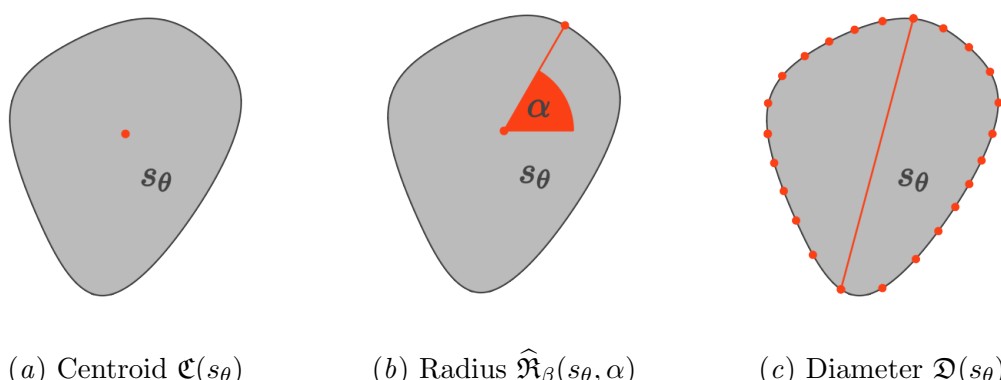

$(a)$ Centroid $\mathfrak{C}(s_\theta)$      $(b)$ Radius $\widehat{\mathfrak{R}}_\beta(s_\theta, \alpha)$      $(c)$ Diameter $\mathfrak{D}(s_\theta)$

Figure 2: Schematic representation of the differentiable module

The centroid and diameter can be learned with a mean squared error, using ground truth centroid $(\frac{l+l'}{2})$ and diameter $(||l - l'||_2)$ as reference:

$$\mathcal{L}_\mathfrak{C}(\theta) = \left|\left|\mathfrak{C}(s_\theta) - \frac{l+l'}{2}\right|\right|_2^2, \tag{8}$$

$$\mathcal{L}_\mathfrak{D}(\theta) = \left|\left|\widehat{\mathfrak{D}}(s_\theta) - ||l - l'||_2\right|\right|_2^2. \tag{9}$$

This provides most of the supervision, but, as we will show in the experiments section, additional regularization is desirable. Imposing a star-shape prior (favoring continuous segmentation with a sharp boundary) is an effective way to do it. This translate into a minimization of the variance of the radius estimate for different $\beta$:

$$\mathcal{R}(\theta) = \mathbb{E}_{\alpha \in \Gamma}\left(\text{Var}_{\beta \in B}\left(\widehat{\mathfrak{R}}_\beta(s_\theta, \alpha)\right)\right). \tag{10}$$

The final combined loss is therefore:

$$\mathcal{L}(\theta) = \lambda\mathcal{L}_\mathfrak{C}(\theta) + \mu\mathcal{L}_\mathfrak{D}(\theta) + \nu\mathcal{R}(\theta), \tag{11}$$

with $\lambda$, $\mu$ and $\nu$ hyper-parameters balancing the different components.

## 4. Experiments

### 4.1. Data

We use the data from the CARE II study (Zhao et al., 2017), where the 24 enrolled patients had a recent ischaemic stroke or transient ischaemic attack. The scans are 3D Motion Sensitized Driven Equilibrium prepared Rapid Gradient Echo, 3D-MERGE. Full lumen segmentations of either left or right internal and common carotid artery are available in average in 12.1% of the slices of a scan. Based on those segmented slices we simulate diameter annotations by picking for each $(l, l')$ such as the points are the furthest away

from each others. This result in 2151 annotated 2D slices over the whole dataset. At training and evaluation time all 2D slices are cut in half to have only one carotid artery per image.

### 4.2. Baselines and ablation study

We compare our method to InExtremIS (Dorent et al., 2021) and CircleNet (Yang et al., 2020), as they are the most relevant and related weakly supervised methods to our problem. To perform a fair comparison across all methods, the additional CRF loss (Tang et al., 2018) used by InExtremeIS is not included; as all methods could benefit from it. We believe that this enables a more exact analysis of each methods performances and limitations. For CircleNet, the input and output spaces have been tweaked to have a similar dimension. This enables us to use the same network architecture for all methods. Additionally, we compared to a fully-supervised U-Net as an upper-bound, and perform an ablation study on the different components of the loss in Eq. (11).

### 4.3. Implementation details

All methods are build on top of the base U-Net architecture (Ronneberger et al., 2015)[4] and are trained with the same ADAM optimizer (e.g. learning rate $10^{-4}$, $\beta_1 = 0.9$, $\beta_2 = 0.99$) for 600 epochs. Our loss components (Equations (8), (9) and (10)) do not require any modification of the network architecture or training regiment, and are implemented as direct losses. We use $|\Gamma| = 24$ radii equally spread between $[0, 2\pi[$ and $B = \{0, 1\}$ as $\beta$ exponents. $\lambda$, $\mu$, $\nu$ and $\sigma$ have been empirically set to 10, 100, 1 and 0.15 experimenting on a private dataset. For all methods the final segmentation is the largest connected component of the network output thresholded at a value 0.5 as having multiple components is not realistic and would result in a systematic over estimation of the diameter. Our publicly available code[5] contains the implementations of both our methods and the different baselines.

### 4.4. Metrics and evaluation

All methods were trained and evaluated over a 4-folds cross-validation (2 folds for training, 1 fold for validation and 1 fold for testing), repeated three times with different random seeds. Evaluation is performed with dice score (DSC), Hausdorff distance[6] (HD) and absolute diameter error (ADE); we report the per-patient average over the testing set (over each run). We also perform a two-sided Wilcoxon signed-rank test, with a level of significance of 0.05 to determine if a run of the cross-validation is significantly different from our method.

## 5. Results

We display the metrics distribution in Figure 3, for each run of the cross-validation. In Table 1, we report the mean (and standard deviation) of the medians of those distributions, computed over the different runs of the cross-validation.

---

4. With the official Monai implementation as starting point: https://docs.monai.io/en/0.7.

5. https://gitlab.com/radiology/aim/carotid-artery-image-analysis/diameter-learning

6. In the case of an empty foreground (no prediction), we set the Hausdorff distance to the diagonal of the image, i.e. 272.4 mm.

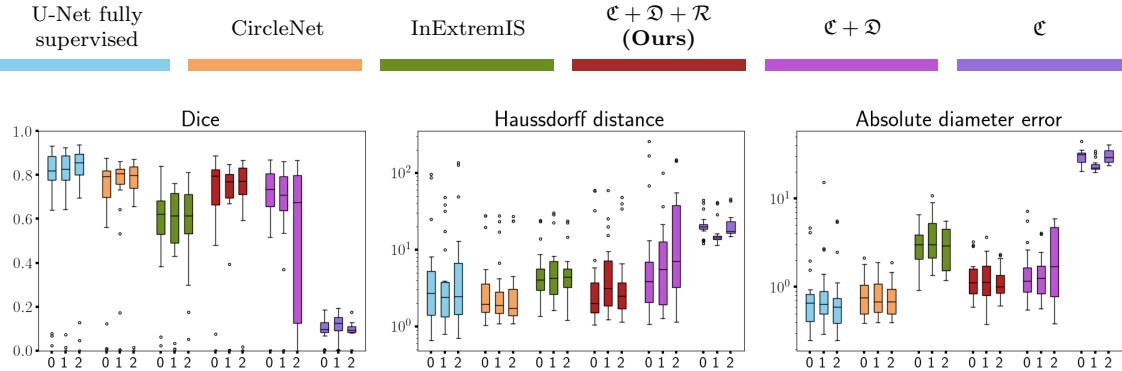

Figure 3: Boxplots of ours metrics, averaged per patient. Each box represents a run of the cross-validation, and go from Q1 to Q3 while displaying the median. The whiskers extends from the box by 1.5 time the inter-quartile range.

Table 1: Mean and standard deviation over the different runs of the medians of the metrics averaged per patient. The superscript highlight runs that are statistically different from our proposed method.

| Method | DSC (%) ↑ | HD (mm) ↓ | ADE (mm) ↓ |
|---|---|---|---|
| U-Net fully supervised | $0.835(0.016)$ [0,1,2] | $2.53(0.13)$ | $0.64(0.03)$ [0,1,2] |
| CircleNet (Yang et al., 2020) | $0.799(0.005)$ [1] | $1.85(0.08)$ [1,2] | $0.71(0.04)$ [0,1,2] |
| InExtremIS (Dorent et al., 2021) | $0.617(0.003)$ [0,1,2] | $4.27(0.16)$ | $2.98(0.02)$ [0,1,2] |
| $\mathfrak{C} + \mathfrak{D} + \mathcal{R}$ (Ours) | $0.780(0.012)$ | $2.54(0.47)$ | $1.09(0.05)$ |
| $\mathfrak{C} + \mathfrak{D}$ | $0.706(0.025)$ [0,1,2] | $5.47(1.29)$ [0,1,2] | $1.38(0.24)$ [1,2] |
| $\mathfrak{C}$ | $0.105(0.015)$ [0,1,2] | $17.14(2.24)$ [0,1,2] | $27.64(3.95)$ [0,1,2] |

We can quickly notice that the best weakly supervised methods (CircleNet and ours) are fairly close to full-supervision performances (3.6% and 5.5% differences, respectively). However, notice that the difference between CircleNet and our method, despite the slightly higher performances for CircleNet (partly explained by the approximative circularity of most segmented arteries, see Appendix D, Figure 6), is not statistically different on each run. The ablation study demonstrates the usefulness of each component of the loss, as removing the regularizer $\mathcal{R}$ induces a 8% drop in performances, while supervising only the centroid $\mathfrak{C}$ gives very poor performances.

Moreover, the qualitative results from Figure 4 highlight a key difference between CircleNet and our method: as CircleNet predict only perfect circles, the segmentation precision decreases when shape-complexity increases (Appendix D, Figure 7). An example is in the second row of Figure 4, where the shape of the carotid in its bifurcation is lost.

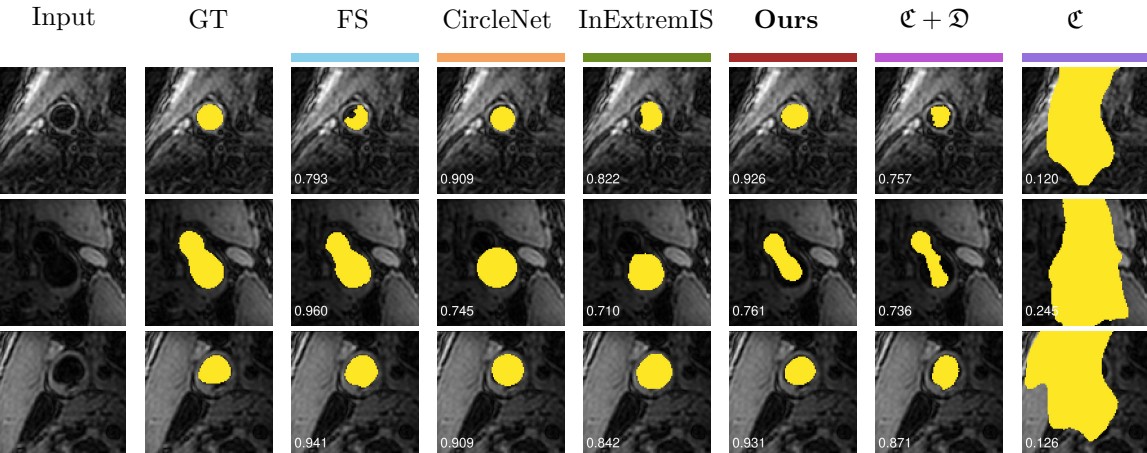

Figure 4: Obtained carotid artery segmentation for the different methods and different examples. The bottom left value indicates the Dice score.

## 6. Discussion

The experiments have shown the strong performances (both quantitatively and qualitatively) of our method, over existing weakly supervised methods. Although CircleNet performed better than our proposed method it does not propose a viable alternative as it does not fully capture the complexity of the shape of the carotid artery bifurcation. However, a more comprehensive evaluation remains to be done. Most notably, we had to simulate the diameter annotations: it would be interesting in a future work to compare clinician annotations to simulated annotations. Indeed, the clinician annotations might not always be *exactly* at the widest diameter of the image, but this slight imprecision might prove (or not) negligible. Another limitation of the current study came from the data heterogeneity: all methods failed in the (under-represented) subset of images ($n = 3$) that had a different spatial resolution.

While the differentiable boundary extraction module in this paper is used for diameter optimization, we are not exploiting it to its full potential: it could be used to model and supervise more complex shapes (Appendix C , Figure 5), such as tumors (Menze et al., 2015). This would be of clinical relevance as the well-established RECIST criterion assesses the progression of tumors based on the evolution of their longest diameter (Schwartz et al., 2016).

## 7. Conclusion

We have introduced a fully differentiable method to locate the centroid and boundary points of an object. This powerful method can then be used to train a neural network, supervising the object shape around its centroid. This provides an elegant way to re-use existing clinical annotations. We validated the method on the task of carotid artery segmentation, using only diameter annotations for training, and showed state of the art weakly supervised segmentation performance.

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

## Appendix A. List of Symbols

| | |
|---|---|
| $u \in \Omega \subset \mathbb{R}^2$ | coordinates in the Cartesian image space; |
| $I : \Omega \to \mathbb{R}$ | input image; |
| $s_\theta : \Omega \to [0, 1]$ | softmax predictions for input $I$ and parameters $\theta$; |
| $(l, l') \in \Omega^2$ | simulated diameter annotations; |
| $\mathfrak{C} : (\Omega \to [0, 1]) \to \Omega$ | function computing the centroid of a (predicted) segmentation; |
| $(r, \gamma) \in \Lambda_\theta = \mathbb{R}^+ \times [0, 2\pi[$ | polar coordinates, centered around $\mathfrak{C}(s_\theta)$; |
| $\Phi_{s_\theta} : \Omega \to \Lambda_\theta$ | function going from Cartesian to polar coordinates; |
| $\alpha \in \Gamma \subset [0, 2\pi[$ | some angle out of a discrete set; |
| $\beta \in B \subset \mathbb{R}^+$ | some exponent out of a discrete set; |
| $\mathfrak{R} : (\Omega \to \{0, 1\}) \times [0, 2\pi[ \to \mathbb{R}^+$ | exact radius of a discrete segmentation; |
| $\widehat{\mathfrak{R}}_\beta : (\Omega \to [0, 1]) \times [0, 2\pi[ \to \mathbb{R}^+$ | approximate radius of continuous probabilities with param $\beta$; |
| $\widehat{\mathfrak{R}} : (\Omega \to [0, 1]) \times [0, 2\pi[ \to \mathbb{R}^+$ | final approximate radius of continuous probabilities; |
| $\widehat{\mathfrak{D}} : (\Omega \to [0, 1]) \to \mathbb{R}^+$ | approximated diameter for continuous probabilities; |
| $(\lambda, \mu, \nu) \in \mathbb{R}^3$ | hyper-parameters balancing the combined loss |

## Appendix B. Radius derivations

Let's derive the integral of Equation (3):

$$\mathcal{I}_\beta(s_\theta, \alpha) = \int_{u \in \Omega} f_{s_\theta, \alpha}(u) s_\theta(u) ||u - \mathfrak{C}(s_\theta)||_2^\beta \, du. \tag{12}$$

We apply the following change of variable: $(r, \gamma) = \Phi_{s_\theta}(u)$

$$\mathcal{I}_\beta(s_\theta, \alpha) = \int \int_{(r,\gamma) \in \Lambda_\theta} f_{s_\theta, \alpha}(\Phi_{s_\theta}^{-1}(r, \gamma)) s_\theta(\Phi_{s_\theta}^{-1}(r, \gamma)) r^\beta r \, dr \, d\gamma \tag{13}$$

$$= \int \int_{(r,\gamma) \in \Lambda_\theta} g_\alpha(\gamma) s_\theta(\Phi_{s_\theta}^{-1}(r, \gamma)) r^{\beta+1} \, dr \, d\gamma \tag{14}$$

$$= \int_{\gamma \in [0, 2\pi]} \int_{r \in \mathbb{R}^+} g_\alpha(\gamma) 1_{x \le \mathfrak{R}(s_\theta, \gamma)}(r) r^{\beta+1} \, dr \, d\gamma \tag{15}$$

$$= \int_{\gamma \in [0, 2\pi]} \int_{r=0}^{\mathfrak{R}(s_\theta, \gamma)} g_\alpha(\gamma) r^{\beta+1} \, dr \, d\gamma \tag{16}$$

$$= \frac{1}{\beta + 2} \int_{\gamma \in [0, 2\pi]} g_\alpha(\gamma) \mathfrak{R}(s_\theta, \gamma)^{\beta+2} \, d\gamma. \tag{17}$$

Under the following assumptions:

$$\int_{\gamma \in \mathbb{R}} g_\alpha(\gamma) \, \mathrm{d}\gamma \approx \int_{\gamma \in [\alpha - 3\sigma, \alpha + 3\sigma]} g_\alpha(\gamma) \, \mathrm{d}\gamma$$
$$\approx 1$$
$$\mathfrak{R}(s_\theta, \gamma) \approx \mathfrak{R}(s_\theta, \alpha) \qquad\qquad\qquad \forall \gamma \in [\alpha - 3\sigma, \alpha + 3\sigma],$$

$$\mathcal{I}_\beta(s_\theta, \alpha) \approx \frac{1}{\beta + 2} \int_{\gamma \in [\alpha - 3\sigma, \alpha + 3\sigma]} g_\alpha(\gamma) \mathfrak{R}(s_\theta, \alpha)^{\beta + 2} \, \mathrm{d}\gamma \tag{18}$$
$$\approx \frac{\mathfrak{R}(s_\theta, \alpha)^{\beta + 2}}{\beta + 2}. \tag{19}$$

Finally we obtain the radius estimate of Equation (5)

$$\widehat{\mathfrak{R}}_\beta(s_\theta, \alpha) = \left( (\beta + 2) \mathcal{I}_\beta(s_\theta, \alpha) \right)^{\frac{1}{\beta + 2}}. \tag{20}$$

## Appendix C. Synthetic examples

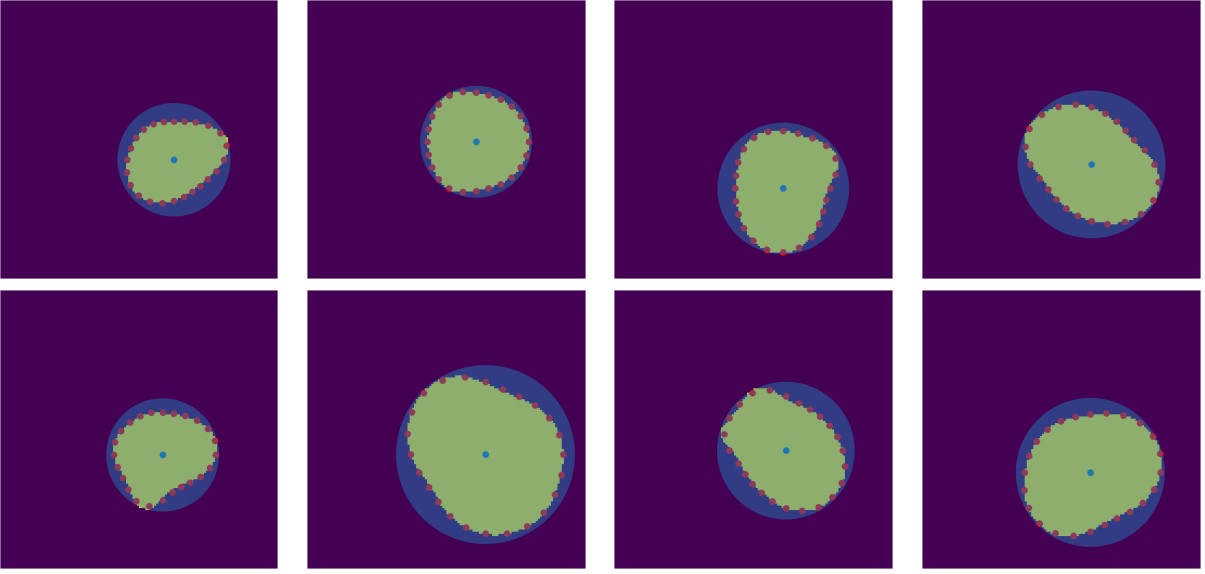

Figure 5: Application of the differentiable boundary point extraction applied to randomly generated star-shaped objects (green: generated star-shape object, red: extracted boundary points, blue: bounding circle).

## Appendix D. Shape analysis

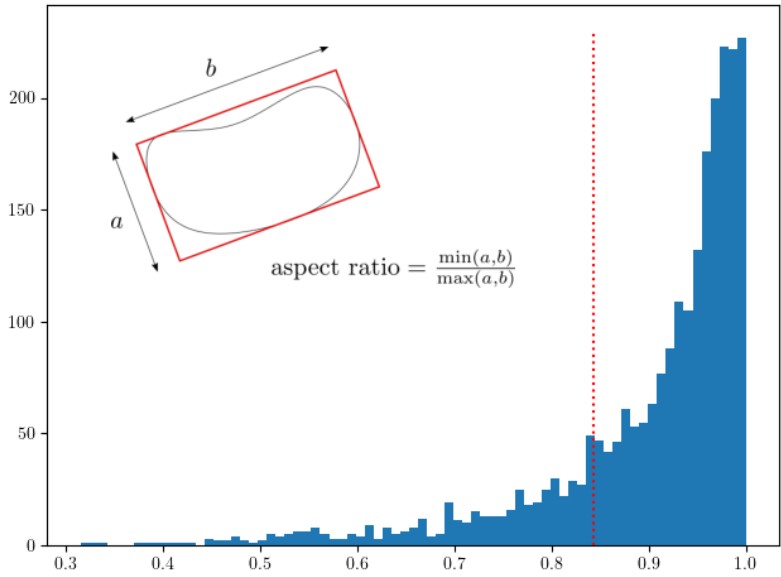

Figure 6: Distribution of the slice-wise aspect ratios of the carotid artery segmentations (the red dotted line represents the $80^{th}$ percentile).

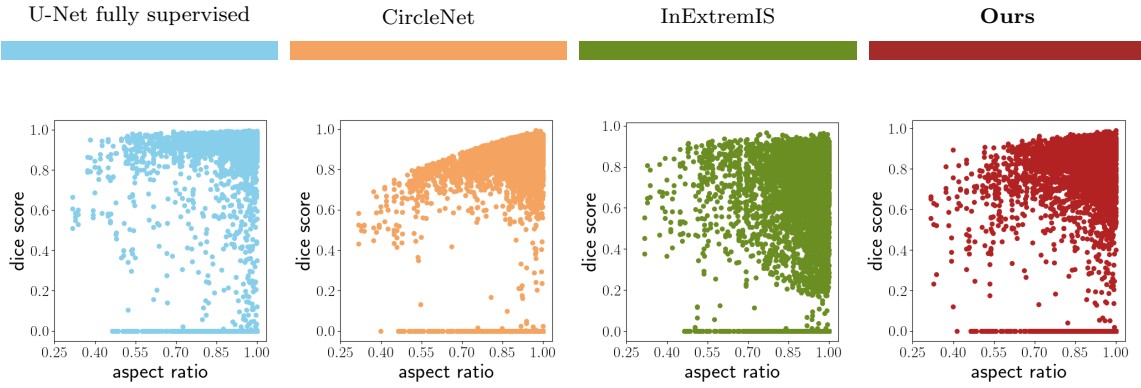

Figure 7: Slice-wise Dice as a function of the slice-wise aspect ratio per method. (Results of all runs of the cross-validation)

