# OpenReview forum: "Differentiable Boundary Point Extraction for Weakly Supervised Star-shaped Object Segmentation"
_MIDL.io/2022/Conference — MIDL 2022_

### Official Review · Reviewer_R11x · 2022-01-05

**Confidence:** 4
**Preliminary Rating:** 4
**Recommendation:** Poster

**Summary:**

This paper present a deep learning module to segment tubular objects from markings of the maximal diameter. The focus of the study is on carotid artery MRI. The key theoretical result is that the weighted average of a distance of a soft-segmentation along a Gaussian arc is a differentiable approximation of the exact radius.

**Strengths:**

* The core theoretical result is clearly presented with precise notation. The derivation is easy to follow and readily interpreted.
* The approach that the authors used to investigate the problem is interesting and clearly driven. It may be useful to generalize this approach to other forms of problems.

**Weaknesses:**

* The extent of the dataset on which the method were evaluated is problematic (only 24 subjects).
* It does not appear that patients were not cross-used between test and validation.
* On the empirical data, CircleNet appears to out perform the proposed method.

**Deanonymize Review:**

no

**Detailed Comments:**

Please double check the formatting of the abstract in OpenReview. I see tex.

**Final Rating After The Rebuttal:**

5: Strong Accept

**Justification Of The Final Rating:**

The authors clearly addressed the limitations outlined. The clarification on separation of training/testing addressed validity concerns. The simulation experiment substantively adds to the validity and novelty.

**Paper Type:**

methodological development

**Questions To Address In The Rebuttal:**

This is a very interesting method and well written paper. However, the results do not justify a substantial advantage over CircleNet. The paper would be much stronger with a clear justification for when the proposed method is most appropriate. For example, is there a simulated imaging scenario in which circlenet would fail, but the proposed module would perform well? Clarity through either discussion or a illustrative example would be very helpful.

**Special Issue:**

no

---

### Official Review · Reviewer_FtjN · 2022-01-23

**Confidence:** 3
**Preliminary Rating:** 4
**Recommendation:** Oral

**Summary:**

The paper proposes a differentiable boundary point extraction method for weakly supervised segmentation.
The proposed method is able to extract boundary points from a model trained using images annotated with only two boundary points required for diameter measurement.
The method is applied on carotid artery segmentation datasets and compared with 2 different methods that can be modified to perform segmentation using only diameter as a label.
The results show that the method achieves state-of-the-art weakly supervised segmentation performance.

**Strengths:**

- The proposed differentiable boundary extraction method is novel and can be quite useful for many medical image segmentation tasks where obtaining pixel-level annotations is cumbersome.
- The results presented in the paper are quite promising in terms of achieving segmentation performance close to SoTA with only 2 point annotation.

**Weaknesses:**

- As also mentioned authors, it would be interesting and useful to extend this method to the segmentation of more complex shapes. However, it is not very intuitive to me how this method extends to more complex shapes with different topologies. I think this should be discussed more, maybe even with some experimental results, since the method is presented as a generic method.

- Although CircleNet only predicts a circle, it performs better compared to the proposed method. This is a bit surprising to me and needs to be discussed more. Is this the case because the majority of the shapes are closer to a circle as in the 1st and 3rd rows in Fig. 4? What are the quantitative scores corresponding to the visual results in Fig. 4?

- There may be different types of weak supervision e.g. bounding boxes, scribbles, etc. How does the method perform compared to methods that use such supervision? I understand that for this specific application, diameter annotation is already done in clinical practice. However, one can still prefer using bounding box annotation if it achieves better results. Therefore, comparisons with different types of weak supervision would be useful.

**Deanonymize Review:**

no

**Final Rating After The Rebuttal:**

5: Strong Accept

**Justification Of The Final Rating:**

I want to thank the authors for spending time to prepare such a greatly detailed rebuttal addressing the concerns/questions of the reviewers. I am very glad to update my rating as strong accept and recommend oral presentation.

**Paper Type:**

methodological development

**Questions To Address In The Rebuttal:**

I found the paper quite interesting and worth publishing in its current form despite some weaknesses. However, I would still be interested in seeing answers for the following questions in rebuttal:
- How does this method extend more complex shapes with different topologies?
- How does it compare with different types of weak supervision?

**Special Issue:**

no

---

### Official Review · Reviewer_tMvV · 2022-01-26

**Confidence:** 4
**Preliminary Rating:** 4
**Recommendation:** Poster

**Summary:**

The paper introduces a new segmentation method. The novelty is to train a neural network to learn from landmarks on the boundary of the segmentation object. Those boundary points come in connected pairs because that is what clinicians routinely annotate. The input data are pairs of landmark points that measure a distance and the output is a 2D segmentation. The authors derive a mathematical equation that is differentiable so that it can be used in a standard neural network framework. They show some preliminary results on real medical images.

**Strengths:**

The paper has a clear structure and very nice graphical illustrations. I like the idea to use existing neural networks and improve them with landmark data that is more affordable than fully segmented images. The preliminary results are promising.

**Weaknesses:**

Here are some points that could be improved in my opinion.

# Introduction

* Your reference to Figure 1 is very late in section 3.2. I would move the Figure there as well.
* Your write: *This type of task-specific weak label has not, to the best of our knowledge, been used in the past.* My suggestion: Could be interesting to compare to conditional statistical shape models:
    * Blanc et al., Statistical model based shape prediction from a combination of direct observations and various surrogates: Application to orthopaedic research, Medical Image Analysis, 2021
    * Blanc et al., Conditional variability of statistical shape models based on surrogate variables, MICCAI, 2009

# Method

* You write: *3.1) locating the centroid of the prediction*. My suggestion: It's not clear what you mean by prediction here.

## 3.2. Boundary points extraction

* I don't understand where the angular coordinate $\gamma$ belongs to. Is this a hyperparameter? Or part of the function $\Phi$?
* What is a differential approximation. Could you elaborate on that? Do you mean the first terms of a Taylor expansion? Also, can you give some intuitions why? Or maybe with some numerical examples?
* Here you use the expectation symbol. I would keep the notation consistent. Either also write the previous integrals with the expectation symbol or change this one to an integral.
* derivable $\to$ differentiable

## 3.4. Losses, regularizer, and final model

* Can you explain equations 8 and 9?
* You write: *This translate into a minimization of the variance*. My question: Only if the expected value is zero, right? Can you relate that to the variance-bias decomposition of the mean squared error? I guess, in your case there is no bias.

# Experiments

## 4.3. Implementation details

* Why an output thresholded of 0.5?

# Discussion

* You write: *Although CircleNet performed better than our proposed method it does not propose a viable alternative as it does not fully capture the complexity of the artery shape.* My suggestion: Then you should use a different metric to show this in your results.


**Deanonymize Review:**

no

**Final Rating After The Rebuttal:**

5: Strong Accept

**Justification Of The Final Rating:**

I would like to thank the authors to responding to all my questions. The authors responded to all my questions adequately. I increased my rating from a weak accept to a strong accept. I congratulate the authors on their work.

**Paper Type:**

methodological development

**Questions To Address In The Rebuttal:**

Points that would improve my rating:

* Clarifying the points I raised in the Method section
* Adding a metric that captures the strength of your method better
* Would it also be possible to include a non-differentiable function and let autodiff handle the differentiation? Would it be worth discussing this in the Discussion section?



**Special Issue:**

no

---

### Official Review · Reviewer_ZgJU · 2022-01-31

**Confidence:** 4
**Preliminary Rating:** 5
**Recommendation:** Poster

**Summary:**

The authors present a weakly-supervised method to harness diameter measurements routinely performed in clinical practice. Starting from a soft map, the authors calculate the centroid. From the centroid over a discrete set of angles, the boundary points are extracted and the diameter measured. Their loss is differentiable, making it directly optimizable through back-propagation. The final loss consists of a centroid, diameter, and regularization part. The results show a consistent performance only slightly lower than a baseline which assumes a circular shape.

**Strengths:**

The paper is clearly motivated by the current clinical practice of using diameter measurements as a surrogate to full three-dimensional segmentation and the scarcity of data for fully supervised segmentation. Prior work is outlined and the research gap clearly carved out. The reader is guided through the derivation of the proposed method and provided with additional information in the appendices. Ablation studies transparently show the contributions of the individual elements. Since the performance is slightly lower than some baseline, the authors make convincing arguments to demonstrate the differences and relevance. I appreciate the clear and instructive writing style.
The code published alongside this submission looks well-documented.

**Weaknesses:**

I have no major weaknesses to report. Some minor remarks:
- The authors compare their method to (among others) CircleNet, which only outputs circular segmentations. I would naively expect that the better case-specific shape of the proposed method is reflected in the Hausdorff and ADE metrics. Are the circular shapes very dominant in the dataset that the non-circular results do not carry enough weight to have an effect?
- With the previous point in mind, are the cases you show in Figure 4 representative?


**Deanonymize Review:**

no

**Detailed Comments:**

No further comments

**Final Rating After The Rebuttal:**

5: Strong Accept

**Justification Of The Final Rating:**

I would like to thank the authors for their detailed responses and clarifications. I found the addition of the artificially generated star-shaped object figure very instructive and the aspect ratio histogram sheds light on the dominance of circular shapes. I recommend accepting this paper.

**Paper Type:**

methodological development

**Questions To Address In The Rebuttal:**

I would appreciate a comment from the authors regarding the discussion of their method's performance better capturing the shape complexity than CircleNet (as also seen in Figure 2) not being represented in the Hausdorff and ADE evaluation.

**Special Issue:**

no

---

### Meta-Review · Area_Chair_EgHv · 2022-02-20

**Recommendation:** Accept (Oral)
**Confidence:** 4

**Metareview:**

The paper proposes a differentiable boundary point extraction method for weakly supervised segmentation. With the weakly-supervised method, the authors harness diameter measurements routinely performed in clinical practice. The novelty is to train a neural network for boundary point extraction given landmarks placed on the boundary of the segmentation object for diameter measurements. Starting from a soft map, the authors calculate the centroid. From the centroid over a discrete set of angles, the boundary points are extracted and the diameter is measured. Their loss is differentiable, making it directly optimizable through back-propagation. The final loss consists of a centroid, diameter, and regularization part.  The method is applied on carotid artery segmentation and compared with 2 different methods
The results show a consistent performance only slightly lower than a baseline which assumes a circular shape.

The reviewers agreed that this is an interesting paper, well-written with nice graphical illustrations, and proposed differentiable boundary extraction method is novel and can be quite useful for many medical image segmentation tasks. They liked the idea of using existing neural networks and improve them with landmark data that is readily available and more affordable than fully segmented images. Further clarifications and figures were added during the rebuttal phase to address some questions raised by the reviewers. As a results all reviewers agreed for a strong accept, and I agree with this decision.

---

### Decision · Program_Chairs · 2022-02-28

Accept